# A Flexible Near-Field Biosensor for Multisite Arterial Blood Flow Detection

**DOI:** 10.3390/s22218389

**Published:** 2022-11-01

**Authors:** Noor Mohammed, Kim Cluff, Mark Sutton, Bernardo Villafana-Ibarra, Benjamin E. Loflin, Jacob L. Griffith, Ryan Becker, Subash Bhandari, Fayez Alruwaili, Jaydip Desai

**Affiliations:** 1Department of Electrical and Computer Engineering, University of Massachusetts Amherst, Amherst, MA 01003, USA; 2Department of Biomedical Engineering, Wichita State University, Wichita, KS 67260, USA; 3Department of Orthopaedic Surgery, Indiana University School of Medicine, Indianapolis, IN 46202, USA; 4J. Crayton Pruitt Family Department of Biomedical Engineering, University of Florida, Gainesville, FL 32611, USA; 5Department of Biomedical Engineering, Duke University, Durham, NC 27708, USA; 6Meinig School of Biomedical Engineering, Cornell University, Ithaca, NY 14850, USA; 7Department of Biomedical Engineering, Rowan University, Glassboro, NJ 08028, USA

**Keywords:** RF resonator, biosensor, microwave sensing, blood flow sensor, hemodynamics, wearable sensor, point-of-care technology

## Abstract

Modern wearable devices show promising results in terms of detecting vital bodily signs from the wrist. However, there remains a considerable need for a device that can conform to the human body’s variable geometry to accurately detect those vital signs and to understand health better. Flexible radio frequency (RF) resonators are well poised to address this need by providing conformable bio-interfaces suitable for different anatomical locations. In this work, we develop a compact wearable RF biosensor that detects multisite hemodynamic events due to pulsatile blood flow through noninvasive tissue–electromagnetic (EM) field interaction. The sensor consists of a skin patch spiral resonator and a wearable transceiver. During resonance, the resonator establishes a strong capacitive coupling with layered dielectric tissues due to impedance matching. Therefore, any variation in the dielectric properties within the near-field of the coupled system will result in field perturbation. This perturbation also results in RF carrier modulation, transduced via a demodulator in the transceiver unit. The main elements of the transceiver consist of a direct digital synthesizer for RF carrier generation and a demodulator unit comprised of a resistive bridge coupled with an envelope detector, a filter, and an amplifier. In this work, we build and study the sensor at the radial artery, thorax, carotid artery, and supraorbital locations of a healthy human subject, which hold clinical significance in evaluating cardiovascular health. The carrier frequency is tuned at the resonance of the spiral resonator, which is 34.5 ± 1.5 MHz. The resulting transient waveforms from the demodulator indicate the presence of hemodynamic events, i.e., systolic upstroke, systolic peak, dicrotic notch, and diastolic downstroke. The preliminary results also confirm the sensor’s ability to detect multisite blood flow events noninvasively on a single wearable platform.

## 1. Introduction

Point-of-care technologies based on flexible radio frequency (RF) resonators are emerging research topics in wearable sensing applications since they offer non-invasive sensing capabilities and higher penetration depth than commonly used optical transducers [1,2,3]. RF resonators are also less likely to respond to unwanted electrical signals experienced in typical electrode-based biopotential recording systems. Furthermore, unlike piezoelectric transducers, RF resonators are immune to electrical fatigue, therefore improving the lifecycle of the sensing system [4]. One of the essential needs of personalized medicine is the ability of the transducer to conform to the uneven dynamic surface of the skin. The conformability of the transducer is also critical for detecting blood flow information from different anatomical locations. Unfortunately, conventional rigid electronics offer significant limitations in the placement of sensors at different epidermal locations due to uneven body surfaces [1,2]. Therefore, a flexible RF resonator can play a vital role in solving this issue by promoting a conformable biointerface [1,2,5,6,7,8,9,10,11,12].

The utilities of nearfield RF resonators have been studied for several decades in the fields of wearable technologies with different transceiver architectures for detecting cardiopulmonary motion, precordial displacements for respiration rate and heartbeat, radial pulse, intraocular pressure as well as intracranial pressure (ICP) [1,2,5,6,7,13,14,15,16,17,18,19,20,21]. However, most applications require rigid antennas as a sensing element or bulky benchtop readout equipment for signal transduction, which lacks epidermal conformance and a wearable form factor. In addition, the prior antenna-based transducers require discrete components for impedance matching, which becomes more challenging in the presence of biological tissue at microwave frequencies [11].

Previous benchmark works have demonstrated miniature readout systems for transducing specific RF attributes such as resonant frequency shift and reflection coefficient modulation. However, those are typically power-intensive solutions since the resonators resonate at very high frequency (VHF) and ultra-high frequency bands (UHF) [13,14,15,16,17,18,19,20,22,23,24], which demand a relatively high power-consuming RF transmitter coupled with a power amplifier. On the other hand, the receiver circuit at microwave frequencies becomes more complex and power-intensive because it requires more filters and low-noise amplifiers to condition the intermediate frequencies to extract the desired demodulated waveform. Furthermore, the readout systems at microwave frequencies also suffer from components limitations in terms of linearity, stability, impedance mismatch, noise figure, bandwidth response, and aging. For example, a readout circuit based on free-running voltage-controlled oscillators with surface acoustic wave filters suffers from the frequency drift of the oscillator due to the aging of components and temperature variation. Prior studies have also reported a wrist pulse detection scheme using a phase-locked loop system and an array resonator with an enhanced sensitivity based on a phase shifter, attenuator, and RF coupler [13,16,17,18]. However, in a practical application, the sensitivity of the readout circuit depends on the subjective calibration of the attenuator and phase shifter, which limits the usability of the sensor on different humans [23].

Considering the existing challenges of skin conformance and limitations of RF components at microwave frequencies; a flexible, compact, and minimally obtrusive system is needed that consumes less energy than the VHF (30 MHz–300 MHz) and UHF (300 MHz–3 GHz) resonator based biosensing system. In this work, we build and investigate a scalable sensing system that includes an adjustable low frequency (near the HF band: 3 MHz to 30 MHz) RF resonator coupled with compact readout hardware that enables multisite hemodynamics detection. Furthermore, the system will consume less power since the RF generation near the HF band requires less energy compared to the higher bands of the VHF and the UHF bands. In other words, the VHF and UHF band transmitters will require more dynamic switching power than an HF band transmitter across the same load effect offered by an RF resonator.

In the proposed system, the sensing element is an open-circuit Archimedean spiral resonator with a coplanar outer loop antenna, chemically etched on a flexible polyimide substrate. The readout circuit consists of (1) a direct digital synthesizer (DDS) for RF generation and (2) a direct conversion receiver comprising a resistive bridge with an envelope detector and an amplifier. The entire system acts as a monostatic sensing system where a single RF transducer is exploited for carrier frequency transmission and modulated carrier reception at the tissue-resonator interface.

Our previous works have demonstrated a modular readout system study based on water and phantom arm studies [1]. In addition, we have also explored the capabilities of RF spiral resonators with different configurations to detect hemodynamics, cardiac output, stroke volume, and intracranial fluid volume by utilizing a benchtop vector network analyzer [2,6,7]. However, the development of the current work showcases an alternative to the traditional benchtop data acquisition system by deploying off-the-shelf components in a wearable form factor. The work also presents experimental waveforms of cardiogenic vibration from the thorax and blood flow from carotid and supraorbital locations, which have not been explored previously using a wearable readout system. In addition to these, the work outlines (including computational and experimental evidence) an improved spiral resonator design leveraging an airgap for enhanced sensitivity to detect multisite limb hemodynamics in upper and peripheral limbs.

## 2. Materials and Methods

The physics of near-field perturbations can be described as the detuning effect of RLC tank circuits at resonance, where the system variables influence the effective permittivity of the capacitively coupled medium, in turn modulating the RF attributes (e.g., S_11_ and phase responses at the resonance band) and the impedances of the resonator [1,5,13,14,16,17,25]. In our chosen application, the coupling capacitance between the resonator and the dielectric tissue under test varies at resonance during field perturbation [1,5,25,26,27]. For example, as blood flows through arteries, it periodically changes the diameter of the artery due to boundary and fluid volume displacement [28]. Thus, the periodic blood flow influences the effective permittivity of the coupled dielectric medium. Consequently, these subtle variations in the effective permittivity and coupling capacitance between the tissue and the resonator result in pulsatile modulation in the RF carrier. 

### 2.1. Sensor Design

A near-field RF resonator, illustrated in Figure 1, can be utilized to detect blood flow from arteries. An open spiral (diameter 30 mm) copper structure with 30 turns (N), 0.35 mm gap width (G), and 0.35 mm trace width (T) was chosen as a near-field radiator and feedback element in the readout system. An essential feature of the resonator is the embedded coplanar loop antenna (Figure 1) which enables power transfer and interrogation to the spiral network through capacitive and inductive coupling [1]. The entire geometry is fabricated on a flexible polyimide substrate with a protective superstrate against air oxidation. The polyimide layers also provide galvanic isolation, enabling capacitive coupling between the resonator and skin.

Unlike our prior works [1,5], the designed structure includes an empirically tuned air gap of 9.4 × 0.93 mm surrounded by 0.94 mm thick PET acrylic tape (Figure 1C). We have found that at around the HF band (30–40 MHz), traditional instrumentations using benchtop equipment and off-the-shelf components cannot register the RF carrier modulation resulting from pulsatile blood flow using the spiral resonator. We hypothesize that the impedance variations due to blood flow at the resonator–tissue interface are tiny enough to be detected and exist within the noise floor of the measuring equipment. In our modified resonator, the inclusion of the air gap adds a variable coupling capacitor at the interface between the resonator and the tissue elements. Therefore, this coupling capacitor becomes a tuning element of the lumped impedance matching circuit when the resonator–tissue system is at resonance. We hypothesize that this coupling capacitance can be influenced significantly due to the change of effective permittivity of the dielectric medium. During blood flow and nearfield perturbation, this added air capacitance results in RF carrier modulation detected by the designed wearable readout system. 

Figure 2b shows a lumped RLC (here, R—resistance, L—inductance, C—capacitance) circuit model of a skin coupled resonator. Here, L1 models the resonator’s impedance in the air, determined from the m1 point, as shown in the smith chart in Figure 2a. The smith chart values were obtained using a single port measurement from a vector network analyzer (Model: R&S ZNC-3). All the impedance measurements were measured concerning 50-ohm port impedance. The impedance-matching network consisting of L2 and C2 approximates the resonator’s S_11_ response over the skin. Figure 2c shows the resonator’s S_11_ response over the skin and from the circuit model described in Figure 2b. The lumped model’s S_11_ response is computed without including the air gap capacitance, C_air. Here, C_air value is not an absolute value in the sensing system. Rather it delineates the effect on the S_11_ parameter of the skin-coupled resonator in the presence of an air gap. From Figure 2c, it is evident that introducing an air gap can shift the band response of the skin-coupled resonator. In practical application, C_air will be modulated by the pulsatile blood flow resulting in RF carrier modulation.

Empirically, we have found that a particular dimension of the air pocket (Figure 1C) for the proposed resonator amplifies the reflected AC variations in the S_11_ and resonant frequency, which can be correlated proportionally to the reflection experienced at the air-skin interface. Since the boundary displacement of the artery and associated tissue deformation is a direct consequence of arterial fluid volume displacement, i.e., blood flow, the modulated signal thus contains vital health information. The field is also being reflected at different layers (interfaces), which gets superimposed with the previous reflections (of different phases) and is considered a gross reflection in the proposed sensing system.

### 2.2. Computational Characterization

Computational characterizations of the resonator were carried out using FEKO, a commercially available full-wave EM solver from Altair (Troy, MI, USA), utilizing the method of moments (MoM) technique. The resonator was modeled in FEKO as a 3D structure based on ideal design parameters and material datasheets utilizing a UFL connector. Figure 3b shows the frequency response of the proposed resonator observed in air and at the radial tissue of a single healthy subject from 30 to 75 MHz, collected using a benchtop vector network analyzer (Model: R&S ZNC-3). The baseline resonance band shifted towards the left due to the increase in capacitance, and the quality factor (i.e., sharpness of the resonant peak) also improved due to the impedance matching with the tissue. The highlighted portion in Figure 3b depicts the periodic modulation of the resonance band due to pulsatile blood flow through the radial artery.

For simulations, the resonator was given material properties of either a perfect electric conductor (PEC) or copper from the FEKO material library (frequency independent with conductivity = 5.813 × 107 S/m) with a simulated conductor depth of 0.015 mm, which is greater than skin depth for the copper material across all simulated frequencies. The PEC resonator was used only for specific absorption rate (SAR) simulations as it tends to provide conservative (overestimating) values. Simulations investigating specific absorption rate (SAR), electric field penetration depth, and electric near-field distributions were run. For SAR and penetration depth simulations, the resonator was simulated over a homogeneous tissue block (Figure 3a) given dielectric properties according to the 2/3 muscle model, whereby the dielectric properties of muscle were obtained for the 20 to 50 MHz frequency range at 10 MHz intervals from [29] are multiplied by a factor of 2/3 as a homogeneous approximation of the dielectric properties of the whole human body [30]. The tissue block dimensions were 20 × 20 × 20 cm for SAR and electric near field distribution simulations and 20 × 20 × 10 cm for penetration depth estimation, with the height of the block perpendicular to the resonator being shorter than the other two dimensions. The clearance between the bottom face of the resonator and the top face of the tissue block was 0.5262 mm (less than the final optimized separation distance of 0.93 mm in Figure 1C). No adhesive PET acrylic tapes were included in the simulations.

The specific absorption rate measures how much power tissue absorbs when exposed to electromagnetic radiation in the radio/microwave wavelengths. It is typically used to gauge the safety of radiating radio and microwave devices. For example, the Federal Communications Commission (FCC) has set SAR limits for the general population at 1.6 W/kg, averaged over any 1 g cube of tissue (assuming a tissue density of 1000 kg/m^3^) for applications on the head or trunk, and 4 W/kg averaged over any 10 g cube of tissue for the hands, wrists, feet, and ankles [1,29,31]. The limits are higher by a factor of 5 for occupational/controlled exposure, that is, 8 W/kg averaged over a 1 g cube for the head and trunk and 20 W/kg averaged over a 10 g cube for the hands, wrists, feet, and ankles. For the UFL connector, which gave higher SAR values than the model of the twisted pair feed, the 1 g and 10 g averages at the resonant frequency of 42.28 MHz were approximately 1.27 and 0.39 W/kg, respectively. Note that these values were simulated using a 6.5 dBm power level, the highest power level used throughout these experiments. SAR values can be scaled directly to the new power level for lower power levels. For example, reducing the power by 1/2 reduces the SAR by the same factor, provided all other simulation parameters remain constant [32].

Near field distribution and penetration depth results at resonance (note that the simulated resonance, 40 MHz, and the actual resonance, ~34 MHz, are numerically different due to the modeling constraints) are displayed in Figure 3c,d, respectively. The penetration depth was found to be approximately 6.8 mm based on the depth beneath the surface of the tissue at which the electric field strength reaches 1/e (~0.37) times that of the electric field strength at the surface of the tissue. Near fields were calculated directly beneath the center of the spiral trace at intervals of 0.05 mm. Two-dimensional normalized electric near field (dB) results are shown in Figure 3c for the front view of the resonator. The near fields are normalized to show the interaction with an embedded simulated artery with a diameter of 2.45 mm at a depth beneath the tissue surface (measured from the center of the artery) of 4.32 mm, intended to replicate a typical radial artery in the systole state at the wrist roughly based on depth, diameter, and expansion measurements performed in [33,34]. The artery was modeled as a dielectric material with properties of blood obtained from [29].

### 2.3. Readout Hardware

The readout hardware combines the features of an RF transceiver, including a programmable AD9851 direct digital synthesizer-based RF transmitter and a receiver unit comprising a resistive Wheatstone bridge coupled with a diode based envelop detector, filters, and amplifier, as illustrated in Figure 4. Here, the impedance bridge circuit is a modified standing wave ratio (SWR) bridge where we are utilizing voltage from a single node instead of two nodes [1,5]. SWR bridges are typically found in many RF instrumentations, e.g., handheld antenna analyzers and scalar network analyzers [1,5]. The resonator is connected to the unknown arm of the impedance bridge (at the coaxial connector in Figure 4), whereas the known arms of the bridge contain 50-ohm resistances. When the resonator is at resonance, it offers a small impedance with a 50-ohm real resistive component. Therefore, the bridge stabilizes at resonance, where the AC voltage difference between the legs becomes minimal. On the other hand, when the resonator impedance is not 50-ohm, the complex load impedance creates an imbalance in the bridge resulting in RF carrier modulation. An envelope detector samples the peak of the modulated RF carrier; after low-pass filtering, the signal becomes proportional to the reflected power from the antenna-resonator system. 

The demodulated signal passes through a diode compensator, shown in Figure 4. The compensation stage recovers the nonlinearities and forward-voltage drop in the diode detector, thus improving the accuracy of the DC-coupled signal. Next, the output signal from the compensator is amplified using a non-inverting amplifier with a theoretical gain of 11. After the first amplification stage, the signal is passed through a passive AC coupler to eliminate the DC component. Finally, the AC-coupled signal is further amplified with a second-stage amplifier. The amplified AC signal contains pulsatile modulation information recorded through an inbuilt 12-bit ADC of an MK20DX256 ARM microcontroller (MCU). The MCU transmits the recorded data to a remote computer via an HC05 Bluetooth module.

The readout system receives power from a 5-volt source which consists of a linear voltage regulator and rechargeable battery. Figure 4A shows the functional block diagram of the complete readout system where the RF stimulus is a 36 MHz, 6.5 dBm continuous sine wave. The choice of frequency is decided by observing the spectral response of the resonator from a benchtop vector network analyzer. Experimentally, it was found that the readout system (Figure 4B) can successfully detect pulsatile blood flow at frequencies ranging from 33.2 to 36 MHz which resides within the resonant band of the tailored resonator.

### 2.4. Methodologies

Since the maximum near field gain occurs at the central axis normal to the resonator plane (Figure 3c), the center was positioned at the palpable sites of the targeted anatomical locations. In the presence of biological tissues, the maximum gain occurs outside the resonator, shifting along the central axis normal to the resonator plane (Figure 3c). The resonator’s center and the targeted artery might have a slight offset during placement on the skin. Note that during data collection from the apex, the subject was under momentary respiratory suspension (i.e., breath-holding following a deep inspiration) for 10 s to eliminate the artifact from the lungs’ movement.

#### 2.4.1. Experimental Setup and Data Acquisition

This work was commenced upon the approval of the Wichita State University Institutional Review Boards (IRBs) to evaluate the device on human subjects. The objective was to investigate whether the proposed instrumentations could capture the encoded details of the hemodynamic events observed during the cardiovascular and systemic blood flow while preserving the characteristic morphology in the demodulated waveforms. The tailored sensor was studied at:Supraorbital location in the supine position;Carotid location in the supine position;Apex location in the supine position;Radial location in the seated position on a healthy male human subject (subject #1, age: 23 years).

The recorded data length was 9 s for each evaluation and saved in a CSV file obtained from an oscilloscope (Model: TBS 1052B-EDU from Tektronix) with a sampling rate of 250 Hz. In addition, the wearable version of the device was studied at a radial location on a different healthy male subject (subject #2, age: 29 years) for 6 s with a sampling frequency of 500 Hz. All the measurements were performed in a lab environment at room temperature.

The data acquisition was conducted in two phases. First, we built and studied the transceiver system as modular circuits and used a benchtop digital storage oscilloscope (DSO, Model: TBS 1052B-EDU from Tektronix) as a data recorder with a sampling rate of 250 Hz, as shown in Figure 5. The oscilloscope data helped us design the amplifier with suitable gain for the wearable readout hardware illustrated in Figure 4B. We also studied the modular system to investigate the multisite hemodynamics obtained from different anatomical locations and set the design parameters, i.e., the gain of the amplifier and the cut-off frequency for the low pass filter stages in the transceiver. Finally, we studied the compact system, as shown in Figure 4B, for radial blood flow detection. During the compact wearable device study, we collected data from a different subject to observe the device’s ability to detect waveforms from various human participants.

#### 2.4.2. Signal Processing

Since the pulse waveforms are vulnerable to noise and artifacts [35], the recorded data from the oscilloscope was processed offline using MATLAB software’s digital signal processing (DSP) system toolbox. The raw time domain responses from the sensing system were treated with a finite impulse response (FIR) hamming window filter with a bandpass frequency of 0.25 to 20 Hz to eliminate the DC drift, high frequencies, and background noises [18]. We also studied the effect of different cut-off frequencies on the recorded radial pulse waveform by varying the low pass end of the bandpass filter to 7.5, 12.5, 25, and 37.5 Hz. Figure 6 shows the bode plot of the designed filters where the phase responses are linear within the passband. Note that the high-pass band transitions of the filters are not visible in Figure 6 since the high-pass cut-off frequency is very small (0.25 Hz), and filters could not achieve the rapid band transition (stop to passband) within the given narrowband, i.e., 0 to 0.25 Hz. The filters’ orders were set to 80 to enable rapid attenuation of the signal after the low pass ends. Further post-processing involved detrending and spectral analysis of the time domain responses through the computation of discrete Fourier transformation using fast Fourier transformation (FFT) to quantify the heart rate, which contains information on sympathetic and parasympathetic nervous activities such as stress response, pulse pressure, and physical activities. Moreover, the intensity of the harmonics in the frequency domain data also contains critical clinical information on myocardial infarction [36].

## 3. Results

### 3.1. Radial Blood Flow Detection

Figure 7a,c demonstrate the raw and filtered radial blood flow responses with reference to an optical photoplethysmogram (PPG) sensor (Part: Pulse Sensor from Adafruit). The peak responses of the two modality signals indicate the leading behavior of the radial pulse concerning the finger pulse, which is obvious since the wrist precedes the fingers anatomically. Moreover, the peak intervals of the two signals (radial and finger pulse) are comparable, revealing the continuous blood flow through the radial artery. Figure 7b shows the AC coupled spectral response of the original radial pulse signal. On the other hand, Figure 7d illustrates the frequency spectrum comparison of the filtered time domain signals from the resonator and the PPG sensor.

The resonator and PPG sensor’s data show that the most prominent frequencies (peak 1 in Figure 7d) are overlapped at 1 Hz. The time domain waveforms from the two different sensing modalities, i.e., optical and the proposed electromagnetic sensor, also exhibit characteristic peaks and notches, which are the consequences of flow separation (backflows) due to arterial resistance variations in the vasculatures [37]. Since the waveforms were obtained at two locations (wrist (radial artery) and finger), the flow resistance could vary accordingly due to different path resistances [37]. As a result, the waveforms will have amplitude and phase differences, i.e., leading and lagging behavior. However, the morphological characteristics, i.e., the presence of peaks and notches, are driven by the same physiological phenomena of the systolic upstroke, systolic peak, dicrotic notch, and diastolic downstroke. It is also evident that the modulation in the resonator’s response is solely due to the periodicity of the radial pulse, excluding any external movement artifact from the subjects’ bodies. Note that, during data collection, the presence of any external body movement was isolated to ensure the fidelity of the acquired signals.

In Figure 7b, the signal-to-noise ratio (SNR) can be determined by defining the noise floor and signal amplitude [35]. Based on our observations between Figure 7b,d, we set the noise floor based on the amplitude of the peak observed at 29.36 MHz (resulting from the RF carrier leakage in the demodulator) and the signal level to the maximum amplitude of the peak (data-1 in Figure 7b) observed at 1 Hz. Based on the amplitude information, the calculated SNR value is 18.18 dB. 

Figure 8 shows the effect of different pass bands on the radial pulse waveform. Filter 1 (passband frequency of 0.25 to 7.5 Hz) attenuates the signal more highly than other filters, whereas filter 4 (passband frequency of 0.25 Hz to 37.5 Hz) includes noise components, as confirmed by Figure 7b. The waveform responses for different low-pass cutoff frequencies also clarify our choice of a bandpass filter with a passband frequency of 0.25 to 20 Hz.

Figure 9 presents a peak-to-peak comparison between the filtered and normalized ppg and the radial pulse waveforms. The time difference between the systolic peaks (point 1 on both waveforms) of the two waveforms is 64 ms, and the diastolic peaks (point 2 on both waveforms) is 128 ms. This pulse transition time information could help determine dynamic blood pressure on the radial artery. Interestingly, the radial pulse waveform shows additional peaks compared to the PPG waveform. These peaks might contain more features from diastolic runoff, which might not be visible in the conventional optical PPG sensor. However, this will remain an important future work to justify our hypothesis based on population study. Furthermore, the normalized amplitude also contains arterial stiffness information expressed by the reflection index.

Finally, Figure 10a shows the frequency spectrum of the radial pulse waveform obtained from subject-2 (age: 29 years) using the wearable sensor (Figure 4B). The data were transmitted to a remote host data acquisition software via Bluetooth telemetry. The sampling rate was 500 Hz. The raw data were processed offline using MATLAB software. Figure 10a shows the applied filter’s response, which largely attenuates the high-frequency noise near 30 Hz. The peak frequency from the raw waveform is 1.5 Hz. Figure 10b shows the radial pulse waveform obtained after applying the FIR filter. The waveforms indicate the presence of hemodynamic events, i.e., systolic upstroke, systolic peak, dicrotic notch, diastolic peak, and diastolic runoff.

### 3.2. Multisite Blood Flow Detection

Similarly, the sensing system was also studied at the apex, carotid, and supraorbital location of a healthy human subject in the supine position. The time domain results are plotted in Figure 11, where the approximate resonator’s sites are labeled with the corresponding recorded waveforms. Figure 11 shows that each waveform contains certain features, i.e., characteristics peaks and notches. Similar waveforms were also reported in prior literature; however, those were obtained either using a benchtop network analyzer, invasive catheterization, or pressure-based transducer during human trials [9,38,39,40]. Hence, we assume that the proposed resonator and the readout system can detect blood flow volume responses from different epidermal locations where palpations or fluid volume displacement occur.

## 4. Discussion

Through this work, we have demonstrated a unique noninvasive sensing modality that can provide users access to the multisite health information on a single sensing platform. To supplement our claims, we have presented a set of data collected from the central (e.g., carotid, supraorbital) and peripheral vasculatures (radial) as well as the thorax (apex position of the heart). Since the supraorbital artery is a terminal branch of the ophthalmic artery which originates from the internal carotid artery and merges with other intracranial arteries to form the circle of Willis; hemodynamics recording from supraorbital and carotid arteries will provide the clinicians with information like dynamic change in intracranial pressure, and intraocular pressure. Furthermore, blood pressure in the superficial temporal artery, e.g., the supraorbital artery, correlates with pressure in the brachial artery [38]. Therefore, our sensing technology could potentially be used as an alternative to a cuffed blood pressure monitor. In addition, the sensor can also be used as a skin temperature monitor and injury prevention detector [41,42].

The characteristic peaks and notches in the pulse waveform are usually the consequence of flow separations and vena contracta effects in the bloodstream due to branching and cross-sectional variation in the arterial tree. According to the vascular dynamics, the ramifications introduce flow resistance, hence eddy separations which further generate reflections in the flow path. These wave reflections could potentially contain information on arteriosclerosis, stenosis, and aorta-to-peripheral pulse pressure amplification, critical parameters in predicting cardiovascular risk [43,44,45,46]. Furthermore, clinical medicine reveals that (through ultrasound velocimetry) a healthy human body exhibits triphasic blood flow responses, i.e., rapid systolic upstroke, early reverse diastolic flow (which results in a dicrotic notch in the pressure waveform), and late forward diastole at peripheral limbs due to higher flow resistance [43]. Although ultrasound velocimetry is an entirely different, as well as expensive, sensing modality in comparison to the proposed near-field epidermal sensor, our previous experimental studies based on detecting bi-phasic and monophasic flow responses from an arm phantom confirmed that the multiphase responses should be registered as a proportional number of peaks in the resonator’s response [1,2]. Accordingly, the experimental results presented in Figure 7, Figure 10, and Figure 11 also ensure the existence of specific characteristic peaks and notches relevant to the hemodynamic events. This confirms that the proposed sensing system predominantly detects the boundary displacement of the organ, hence pulse pressure from the arteries.

Specific unique signatures are also evident from the multisite hemodynamics study presented in Figure 11. For example, the transduced waveform from the apex might reflect the displacement cardiogram (DCG). In addition, DCG provides information on cardiac timing intervals, such as the pre-ejection period (PEP) and left ventricular ejection time (LVET) [26,47]. Therefore, we believe that the features in the waveform exclusively contain clinically significant hemodynamic information. However, these results are still open to clinical interpretations requiring comprehensive multisite hemodynamics studies using electromagnetic sensing technology, which will remain an important future work.

In the current configuration, motion artifacts and other dielectric interferences can corrupt the pulse signals since the resonator utilizes a mechanical UFL connector to communicate with the readout circuit. In addition, the simulation results shown in Figure 3c reveal the maximum gain and directivity of the near-field at a point external to the resonator-skin interface. Therefore, any dielectric interference in the maximum gain areas could significantly corrupt the signal, e.g., adding DC drifts in the blood flow modulated waveforms. However, the issue can be solved by adding an RC average circuit and a differential amplifier stage or a switched capacitor in the demodulator unit of the detection circuit, which will be investigated in future iterations of the proposed sensor.

We also studied the effect of different low-pass cut-off frequencies on the radial pulse waveform and found pronounced attenuation of the signal for a cut-off frequency of 7.5 Hz. Prior studies have reported bandpass filters where the corner frequencies of the low pass end range from 15 to 20 Hz for PPG waveform analysis [18,35]. Studies have also shown that different corner frequencies potentially influence the waveform morphologies, especially the location of the feature points [48]. However, we did not observe any significant differences from our dataset reported in Figure 8 concerning the waveform morphology. From Figure 6, the designed window filters did not achieve the desired band transition at the high-pass end due to the narrow filter bandwidth of 0 to 0.25 Hz. This effect did not alter the waveform morphology in our data set compared to the raw time domain data. However, the resulting filter might not possibly eliminate the DC drift in the data set. This issue can be solved by constructing the high-pass and low-pass filters separately and processing the raw time domain data accordingly [48]. In addition, a study has reported that a pulse waveform longer than 10 s can provide respiratory influences from different body sites [49], which will remain an important future work for the multisite evaluation of the sensing system.

Furthermore, the overlapping vasculatures and the sensor misplacement might also influence the response. A relatively smaller resonator could potentially mitigate this problem since the smaller form factor would increase the radar cross-sectional area of the arteries. Nevertheless, the smaller the resonator, the higher the resonant frequency, which might add hardware challenges to the point-of-care version of this technology. Moreover, microwave circuits require careful impedance routing and shielding to reduce reflections at the RF components and carrier modulation due to unwanted positive feedback at the amplifiers in the transceiver. 

In addition, from Figure 3b, it is also evident that the pulsatile blood flow also modulates the resonance band. Therefore, the resonant band might be subjected to change since the dielectric condition of human skin depends on multiple variables, including hydration status and chemical compounds on the skin. Hence, the sensing system also demands narrowband ultra-fast sweeping instead of single-frequency interrogation. Finally, a future iteration of this technology is looking forward to utilizing very large-scale integration (VLSI) technology to develop a dedicated application-specific integrated circuit (ASIC) to enable a true skin patch sensor.

## 5. Conclusions

Herein, we have presented a wearable sensing system for measuring multisite limb hemodynamics. Enabled by a compact wearable readout design, this technology offers the potential for monitoring blood flow from peripheral and central vasculature at minimal cost to facilitate point-of-care diagnosis. Using this technology, we obtained radial pulse waveforms from two human subjects with different heart rates, i.e., 60 beats per minute (bpm) and 89 bpm. These results showcase the sensor’s ability to distinguish between two different heart rates and waveform morphologies. In addition, we obtained a systolic peak-to-peak transition time of 64 milliseconds and a diastolic peak-to-peak transition time of 128 milliseconds from subject #1 with a heart of 60 bpm. This pulse transit time could potentially correlate with the dynamic arterial pulse pressure. Therefore, the proposed technology could be used as a cuffless blood pressure measuring system, saving arteries from tremendous mechanical stress. In addition, we have observed multiple peaks in the radial pulse waveform compared with the PPG ground truth. These additional peaks might reveal more clinical information on arterial health. To the best of our knowledge, this is a first-of-its-kind noninvasive epidermal device that has demonstrated the possibility of detecting hemodynamics information from different anatomical locations (clinically significant) utilizing a flexible low-frequency resonator at 34.5 ± 1.5 MHz. Furthermore, the preliminary human trials demonstrated the sensor’s ability to distinguish site-specific hemodynamic information. Future studies will emphasize the clinical evaluation of this sensing technology to promote access to multisite health information on a single wearable platform.

## Figures and Tables

**Figure 1 sensors-22-08389-f001:**
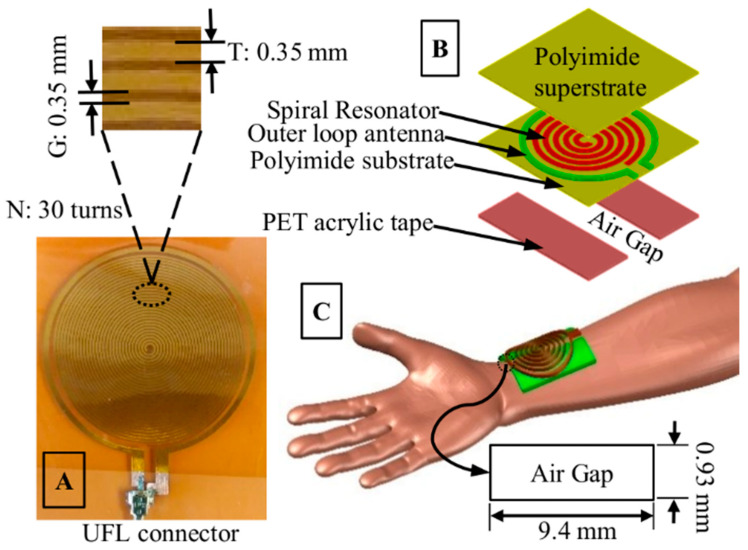
The proposed resonator design illustrating (**A**) the number of turns, N = 30, trace width, T = 0.35 mm, gap width, G = 0.35 mm, and diameter, d = 30 mm of the planar spiral structure. (**B**) shows the exploded view of the resonator, and (**C**) shows the placement of the sensor on the radial location with approximate air gap dimension.

**Figure 2 sensors-22-08389-f002:**
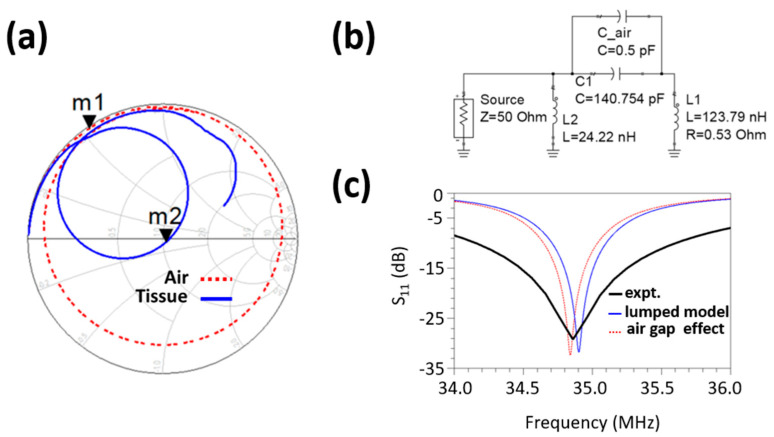
(**a**) Smith chart showing the complex S_11_ plot of the resonator with respect to the air and over skin at radial location from 1 MHz to 100 MHz carrier frequency, (**b**) lumped RLC circuit model of the spiral resonator coupled with tissue and air gap capacitance, and (**c**) comparison of the S_11_ magnitude plot resulting from the lumped model to showcase the probable effect of air gap at the resonator tissue interface.

**Figure 3 sensors-22-08389-f003:**
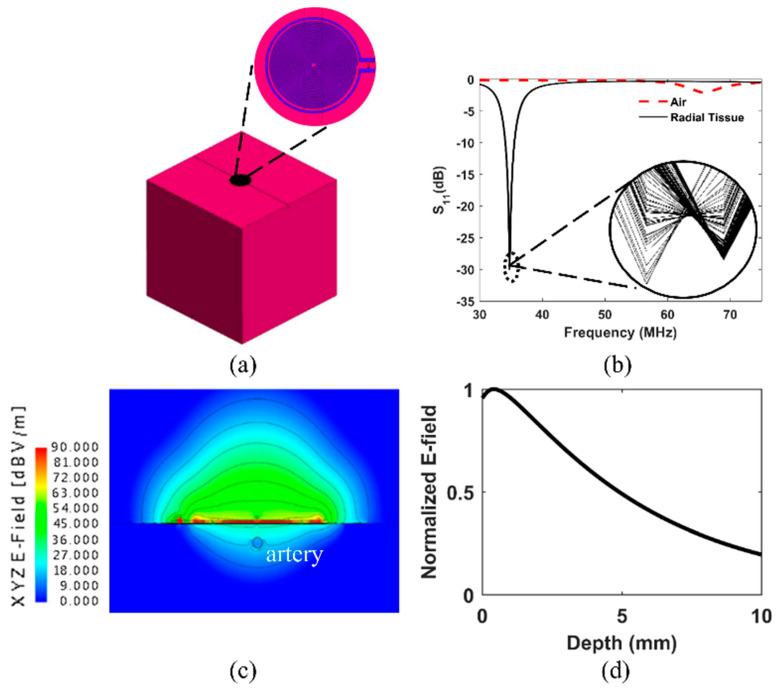
(**a**) A FEKO model of the spiral resonator and 20 × 20 × 20 cm tissue block with an embedded radial artery and surface deformation simulating the systole state. (**b**) Spectral response of the proposed resonator observed at air and radial tissue of a single healthy subject from 30 to 75 MHz- collected using a benchtop vector network analyzer. (**c**) Normalized electric near field (E-field) distribution for the resonator at a resonant frequency near 40 MHz. The resonator is centered over a 20 × 20 × 20 cm tissue block with an embedded artery and small semi-circular deformation at the tissue’s surface during the simulated systole state. (**d**) Normalized E-field strength plotted against approximate depth beneath the tissue block’s surface (zero depth on the plot corresponds to 0.05 mm beneath the tissue surface). The penetration depth is when the normalized field strength reaches approximately 0.37 times the strength at the surface of the tissue.

**Figure 4 sensors-22-08389-f004:**
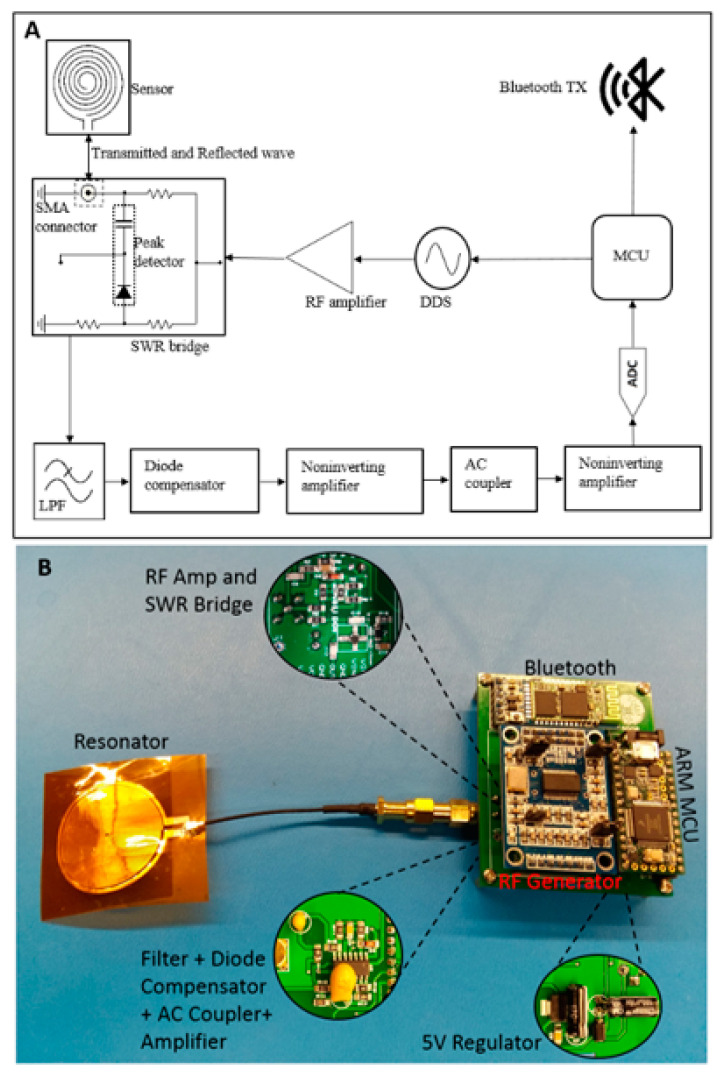
(**A**) Functional block diagram of the proposed sensing system, and (**B**) the developed wearable prototype showing the spiral resonator and readout hardware.

**Figure 5 sensors-22-08389-f005:**
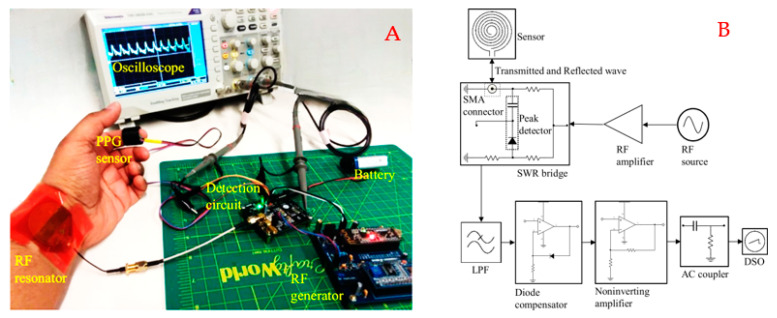
(**A**) Experimental setup for radial pulse detection, (**B**) block diagram of the recording system (contains the RF transmitter and the demodulator circuits) based on modular detection circuits with a benchtop oscilloscope to record the demodulated signal.

**Figure 6 sensors-22-08389-f006:**
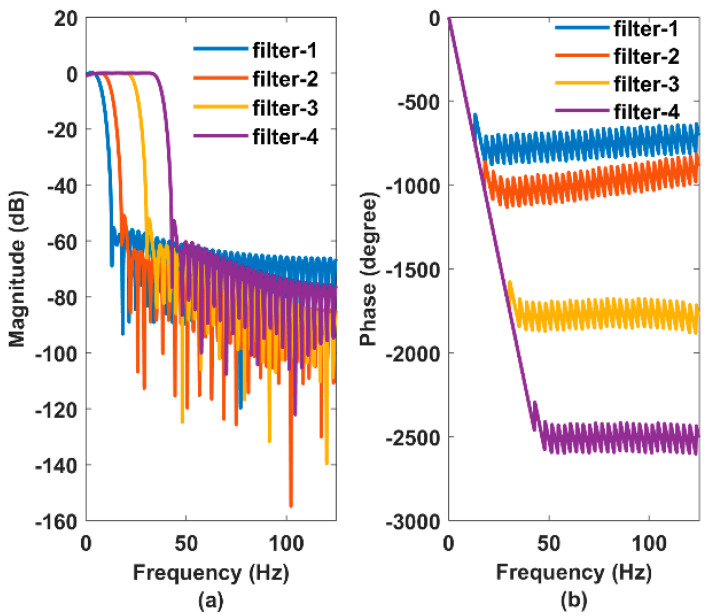
Bode plot of the 80th order FIR Hamming window filters where (**a**) shows the magnitude responses and (**b**) shows the phase (unwrapped) responses.

**Figure 7 sensors-22-08389-f007:**
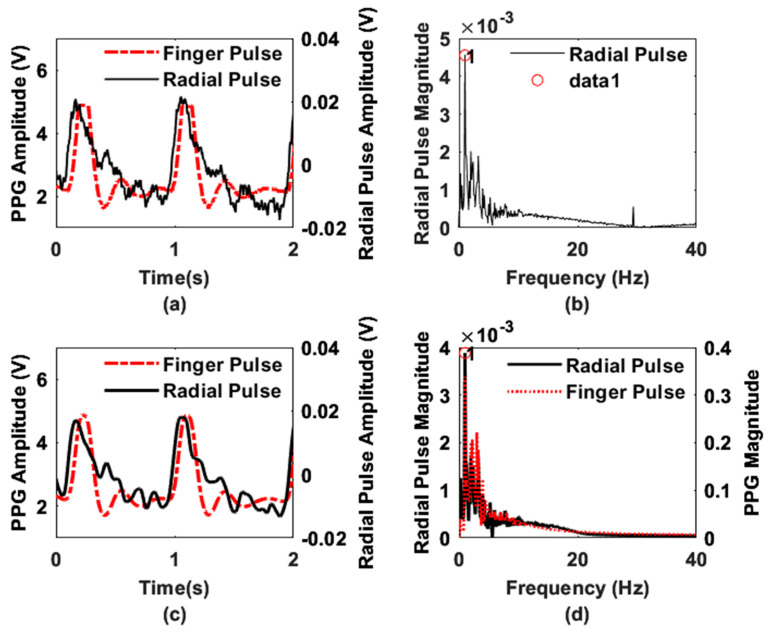
Demodulated waveform obtained for 36 MHz RF carrier during pulsatile blood flow at the radial location of a human subject. Here, (**a**) raw (original) and (**c**) filtered time domain signals and (**b**,**d**) the frequency spectrum of the raw and filtered signals, respectively. In (**b**) data 1 represents the peak of the maximum amplitude.

**Figure 8 sensors-22-08389-f008:**
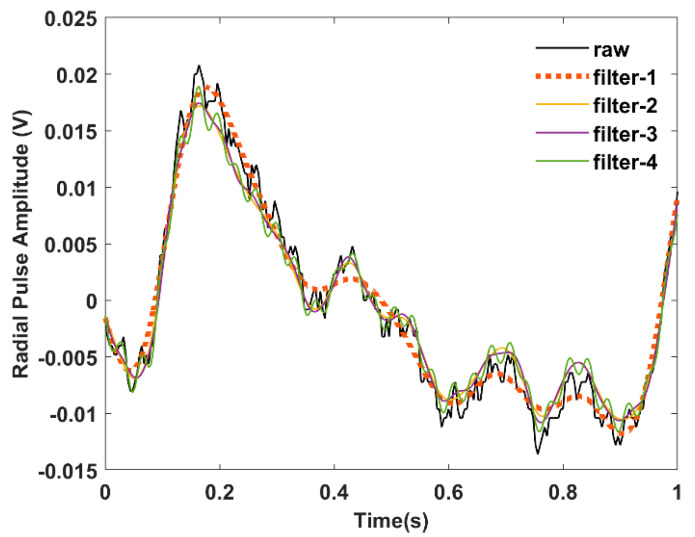
Effect of different low-pass cut-off frequencies on radial pulse waveform.

**Figure 9 sensors-22-08389-f009:**
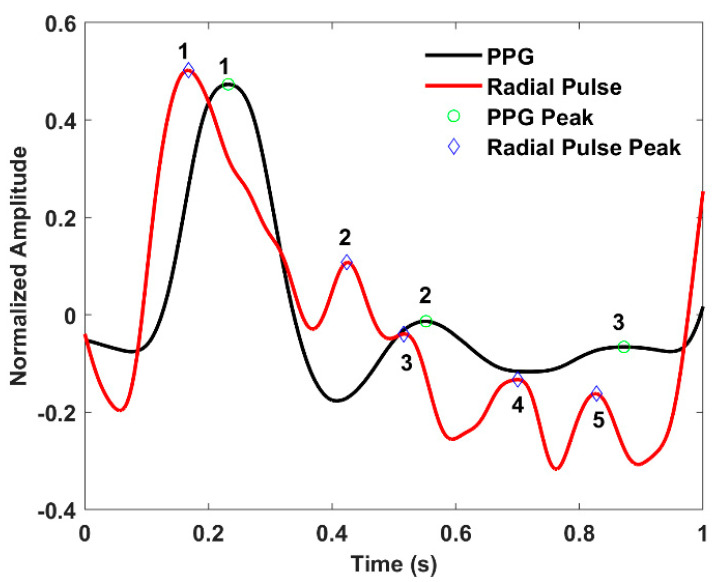
Comparison of the filtered PPG and the radial pulse waveform highlighting the number of peaks.

**Figure 10 sensors-22-08389-f010:**
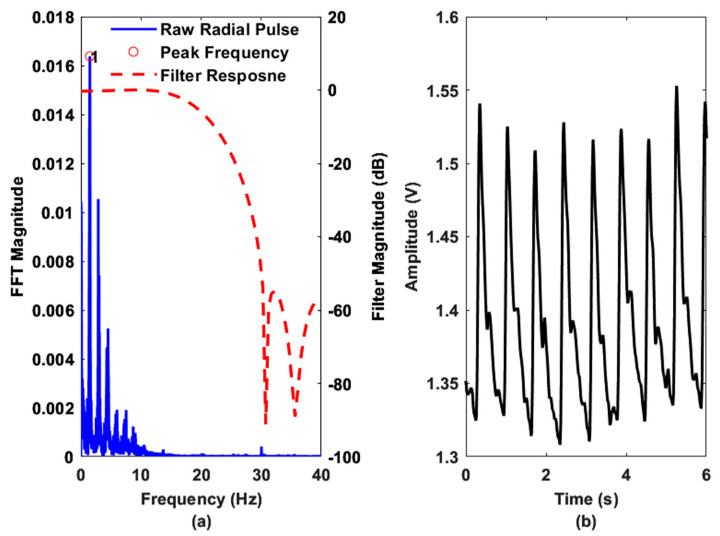
Radial pulse data obtained from the designed compact wearable sensor via Bluetooth telemetry. Here, (**a**) shows the frequency spectrum of the raw radial pulse data along with the applied filter response and (**b**) shows the filtered pulse waveform in time domain.

**Figure 11 sensors-22-08389-f011:**
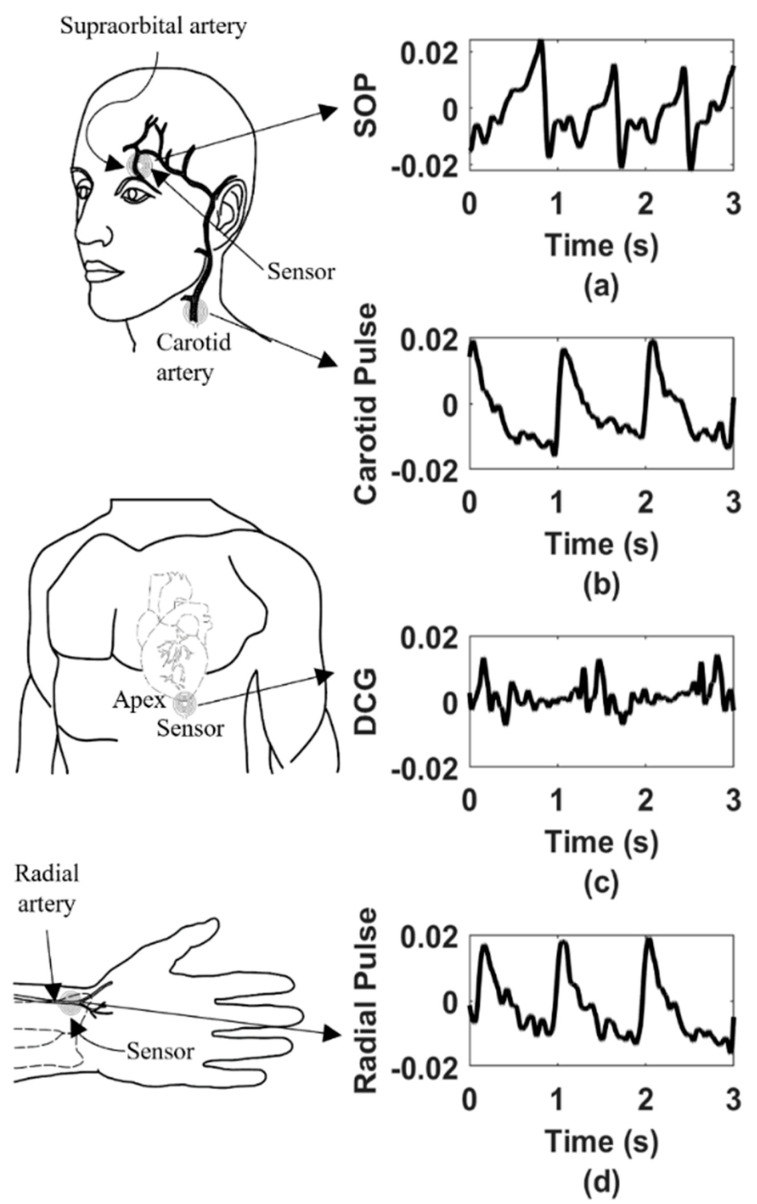
Multisite pulse volume response obtained from a healthy adult male due to the hemodynamic activities observed at (**a**) supraorbital (where SOP means supra orbital pulse), (**b**) carotid, (**c**) thorax (apex site where DCG stands for displacement cardiogram), and (**d**) radial locations.

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
