# Peer review of "A Flexible Near-Field Biosensor for Multisite Arterial Blood Flow Detection"

_sensors, 2022, doi:10.3390/s22218389_

Round 1
Reviewer 1 Report
In this study, the authors developed a wearable near-field biosensor for multisite arterial blood flow detection. The paper is well written and structured. However, some concerns of methodological need to be addressed to meet the publishable standards.
1. "The raw time domain responses from the resonator were treated with a finite impulse response (FIR) bandpass filter having a cut-off frequency of 0.1 Hz to 10 Hz to eliminate the DC drift, high frequencies (>10 Hz), and background noises.” 10Hz is a low cutoff frequency for pulse wave measurement. The filtering parameters can significantly change the pulse waveform and locations of feature points (Refer: 10.1088/1361-6579/ac0a34). Please provide the key parameters to the FIR bandpass filter in the methods, and add the discussion on the possible filtering effect on pulse wave.
2. Please provide more details on the subject information and measurement protocol, e.g., how many subjects were recruited, mean and SD of age, how long was the measurement, in what posture, etc.
3. For a pulse wave longer than 10s, the modulation effects on the signal can be significant in amplitude, baseline, and frequency (Refer: 10.3389/fphys.2019.00732). It will be interesting to compare the intensity of respiratory modulations in the pulse waves from different body sites.
Reviewer 2 Report
This manuscript develops a flexible near-field biosensor for multisite arterial blood flow detection, capable of non-invasively detecting multisite hemodynamic information, which positively contributes to the field of wearable assessment for cardiovascular health. However, some issues are not clear enough, so I think this manuscript needs some revision.
The main problems are as follows:
1. Abstract Line 9 The author should state what Flexible RF is shorthand for.
2. Keywords The author should reduce the Keywords a bit.
3. Introduction Line 41 The author should adjust the format.
4. Section 2.3 The authors should provide additional information on the specific procedure of the experiment, including the acquisition posture, number of subjects, etc. In addition, whether the resonant frequencies used are generalizable.
5. Section 2.3.2 Line 298-300 The author should present more sympathetic parasympathetic information in the frequency domain in section 3 to compare with the PPG and verify the reliability of the system acquisition data.
6. In Section3 the author should provide the actual power consumption of the system to verify the accuracy of your hypothesis.
7. Section 3.1 Could the authors consider additional methods to verify system reliability, such as those suggested in comments 5. Certainly, SNR is another approach that could be considered.
8. Section 3.2 The authors only verified the feasibility of the acquisition and did not evaluate it, which is not the same as the statement in Abstract Line 21-23. Other methods could be considered to assess the reliability of data acquisition.
9. Section 4 Line 371-373 Authors should add some references.
Reviewer 3 Report
The authors designed a radial antenna based flexible biosensor for Multisite Arterial Blood Flow detection.
They also designed a transmitter and receiver circuit to make the measurement using a sensor. The article considers a topical and important issue.
Comment #1
Although the subject of the article is important and well studied, there is one aspect that I think is important.
It is seen that the proposed system and sensor antenna were already designed by one of the authors, Noor Mohammed, in the master thesis titled "A NONINVASIVE, ELECTROMAGNETIC, EPIDERMAL SENSING SYSTEM FOR BIOFLUID PHENOMENA DETECTION" published in 2019 (https://soar.wichita.edu/bitstream/handle/10057/17125/t19063_Mohammed.pdf?sequence=1).
In addition, the results in Figures 5, 6 and 7 given in the article were published in 2019 in the same thesis. When we consider the Ithenticate report, there is usually about 20% similarity with their own publications.
In addition, antenna design and electronic system designs used in the previous publication "A Noninvasive, Electromagnetic, Epidermal Sensing Device for Hemodynamics Monitoring" and the thesis titled "A NONINVASIVE, ELECTROMAGNETIC, EPIDERMAL SENSING SYSTEM FOR BIOFLUID PHENOMENA DETECTION" were used exactly or with minor changes by the authors.
Novality should be increased according to the authors' own previous publications.
Comment #2
The authors did not mention numerical results in the Conclusion section. The Conclusion section is simple and short. In the Conclusion section, they did not discuss what was done in the study and the results obtained.
Comment #3
The authors self-cited 5 previous studies ([1, 5, 13, 14, 16, 17, 25], Line 127), however, it is seen that the publications [13, 14, 16, 17, 25] belong to other authors. Reference numbers should be checked.
Comment #4
There are some typos. It should be checked for spelling errors and grammar.
Round 2
Reviewer 1 Report
Thanks for the update. My earlier comments have been well addressed. I advise the authors to double check the format and find a native English speaker to proofread the manuscript.
Author Response
Thanks for your valuable comments. We have thoroughly revised the manuscripts for possible typos and mechanical errors and improved the readability of the overall manuscripts. Please find the revisions in our revised manuscript.
Reviewer 2 Report
The authors have responded well to my comments and the current edition is ready for publication.
Author Response
Thanks for your valuable comments to improve the manuscript's technical merit and readability.
Reviewer 3 Report
Thank you for taking my comments into consideration. A minor correction to the revised version is required.
Minor issue:
- Figure 3 number is duplicated. All figure captions and references to figures in the text should be checked.
Author Response
Thanks for your valuable comments. We apologize for our mistakes. We have revised the manuscripts for possible typos and mechanical errors and corrected the figures' reference numbers.